# Caffeic Acid Phenethyl Ester Encapsulated in Self-Assemble Rice Peptides Nanoparticles: Storage Stability, In Vitro Release, and Their Interaction Mechanisms

**DOI:** 10.3390/foods13050755

**Published:** 2024-02-29

**Authors:** Xinyue Wang, Siyi Feng, Hongdong Song

**Affiliations:** 1School of Health Science and Engineering, University of Shanghai for Science and Technology, Shanghai 200093, China; wangxinyue20210905@163.com (X.W.); siyifeng0610@163.com (S.F.); 2National Grain Industry (Urban Grain and Oil Security) Technology Innovation Center, Shanghai 200093, China

**Keywords:** rice peptides, self-assembly, CAPE, water solubility, fluorescence quenching

## Abstract

Caffeic acid phenethyl ester (CAPE) is an important active component of propolis with many bioactivities. However, its efficiency and practical application are restricted due to its poor aqueous solubility and storage stability. In this study, a nanocarrier was fabricated to encapsulate CAPE using self-assembled rice peptides obtained by controllable enzymolysis. The physicochemical properties, encapsulation efficiency, and loading capacity of rice peptides nanoparticles (RPNs) were characterized. The storage stability, in vitro release, and interaction mechanisms between CAPE and RPNs were investigated. The results showed that RPNs, mainly assembled by disulfide bonds and hydrogen bonds, possessed an effective diameter of around 210 nm and a high encapsulation efficiency (77.77%) and loading capacity (3.89%). Importantly, the water solubility of CAPE was increased by 45 times after RPNs encapsulation. Moreover, RPNs encapsulation also significantly increased CAPE stability, about 1.4-fold higher than that of unencapsulated CAPE after 18-day storage. An in vitro release study demonstrated that RPNs could delay the release of CAPE, implying a better CAPE protection against extreme environments during digestion. Hydrogen bond and van der Waals force are the predominant interaction forces between RPNs and CAPE. Therefore, the newly developed nanoparticle is a potential delivery system that could effectively improve the aqueous solubility and stability of CAPE.

## 1. Introduction

Caffeic acid phenethyl ester (CAPE) is one of the main biologically active components in propolis [1]. It is known for its excellent biological activities, including anticancer, anti-inflammatory, antibacterial, and antioxidation [1,2]. Especially, CAPE has been demonstrated to have significant cytotoxicity on tumors and carcinoma cells such as the colorectal cancer (CRC) cell line, breast cancer cell line and human leukemic cell line (HL-60), but has very low toxicity compared to normal cells [3]. With the increasing emphasis on human health, CAPE will have broad application prospects in the fields of food and medicine. However, the low water solubility and poor stability of CAPE greatly limits its practical application. The benzene ring and acrylate structure of CAPE determines its low water solubility (<10 μg/mL) [1,4]. Moreover, the phenolic hydroxyl groups on the benzene ring are easily oxidized, which makes CAPE have poor stability. Therefore, it is necessary to develop an efficient strategy to improve the water solubility and stability of CAPE for its better utilization.

The application of micro/nano-encapsulation technology has become science hotspots in the fields of food, nutrition, medicine, etc. It can protect bioactive compounds against chemical degradation, increase their water solubility, and achieve a controlled or/and targeted release. Several micro or nanoencapsulation systems have been reported for improving the water solubility and stability of CAPE. Hydroxypropyl-β-cyclodextrin (HP-β-CD) has been utilized for CAPE encapsulation by forming inclusion complexes with an increased solubility of about 800 μg/mL [5], but this synthetic polymer has not been approved in the study field of the food industry. Loading CAPE in composite nanoparticles such as phosphorylated walnut protein/chitosan (PWPI/CS) nanocomplexes and Ca-casein-NaAlg nanoparticles could effectively reduce the release of encapsulated CAPE and prevent digestion of CAPE [4,6]. However, the utilization of large amounts of encapsulation materials could lead to a decreased loading capacity and organ toxicity during the metabolism [7]. Meanwhile, the interaction mechanism between CAPE and the carrier materials was not clarified in previous studies.

In recent years, food protein-based nano-delivery systems have attracted increasing attention because of their advantages in biocompatibility, nutritional value, and self-assembly capability [8]. Among all food proteins, rice protein has aroused great interest of researchers because of its unique nutritional value, massive production, hypoallergenic property, sustainability, and health reasons. However, the high content of hydrophobic amino acids in rice protein leads to a poor water solubility, which limits its application in the field of nano-delivery. Enzymatic hydrolysis is an effective means to obtain amphipathic peptides, which can be self-assembled into stable nanomicelles or nanoparticle systems [9,10]. Meanwhile, the obtained peptides or hydrolysates with enzymatic hydrolysis of rice protein showed higher biological activities such as anti-oxidation, anti-obesity, and anti-cancer activities [11]. Therefore, enzymatic hydrolysis of rice protein provides a feasible strategy to fabricate nanoparticles for CAPE encapsulation.

In the current study, rice protein was controlled and hydrolyzed, and the obtained self-assembled rice peptides were utilized to fabricate nanoparticles. The physicochemical properties, including the encapsulation efficiency, and loading capacity of rice peptides nanoparticles (RPNs), were characterized. Thereafter, the storage stability and in vitro release of RPNs were evaluated. Finally, the interaction mechanisms between CAPE and RPNs were investigated.

## 2. Materials and Methods

### 2.1. Materials

Milled rice grains were provided by Bright Agricultural Development Group (Shanghai, China). Caffeic acid phenylethyl ester (CAPE) with a purity of 97% was purchased from Aladdin (Shanghai, China). Protease from *Bacillus licheniformis* (P4860) and digestive enzymes including pepsin from porcine gastric mucosa (P7000) and Pancreatin from porcine pancreas (8 × USP specifications, P7545) were provided by Sigma-Aldrich (Shanghai, China). Other chemicals including hydrochloric acid, sodium hydroxide, sodium dodecyl sulfonate, sodium bicarbonate, urea, and dithiothreitol were of analytical grade and purchased from Sinopharm Company (Shanghai, China).

### 2.2. Preparation of Rice Protein and Peptides

Rice protein was extracted according to a previous study [12] with some modifications. Briefly, the rice sample was grinded into powders and degreased using normal hexane. The degreased rice powders were dispersed in deionized water (1:10, *w*:*v*) and the pH was adjusted to 12 using 1 M NaOH. The extraction was carried out for 1 h at room temperature, followed by being centrifugated at 3000× *g* for 10 min. The pH of the obtained supernatant was adjusted to 5 using 1 M hydrochloric acid and then was centrifugated at 3000× *g* for 10 min. The obtained precipitate was washed and freeze dried to obtain the rice protein isolate. The Kjeldahl method was used to analyze the total nitrogen content of protein isolate. A nitrogen conversion factor of 5.95 was used to determine the total protein content of the rice protein isolate [13]. The protein content of rice protein was determined to be about 92%. To prepare the rice peptides, rice protein (20 g/L, pH 8.0) was mixed with Alcalase^®^ 2.4 L (1:100, *w*/*w*, enzyme to substrate). The enzymolysis was performed at 50 °C for 1.5 h, allowing for the controlled enzymolysis of rice proteins. The self-assembled peptides were obtained after hydrolysis with a degree of hydrolysis of about 18%. A total of 4 g/L rice peptides could self-assemble into nanoparticles in deionized water.

### 2.3. Particle Size and Polydispersity Index (PDI) Analysis

A dynamic light scattering (DLS) instrument (NanoBrook 173Plus, Brookhaven Instruments, Holtsville, NY, USA) was used to analyze the size and PDI of the self-assembled rice peptides nanoparticles (RPNPs). A total of 2.5 mL of samples with a concentration of 1 mg/mL were dispersed in deionized water. The incident angle of light was 90°, and the refractive index of water was set to 1.333. All measurements were performed with three replicates at 25 °C.

### 2.4. Analysis of Internal Forces of RPNPs

The internal forces of RPNPs were analyzed based on a previous study [14]. In brief, RPNPs samples were dispersed in different perturbing solvents including distilled water, 0.5% SDS, 30 mM DTT, and 6 M urea alone or in combination. The particle size of RPNPs in different perturbing solvents was analyzed with a particle size analyzer (NanoBrook 173Plus, Brookhaven Instruments, Holtsville, NY, USA). The interactive forces that maintained the structure of RPNPs were analyzed according to the size change in the nanoparticles in different perturbing solvents.

### 2.5. Encapsulation of CAPE by Rice Peptides

Rice peptides with a concentration of 4 mg/mL were mixed with CAPE (10 g/L ethanol solution) with a volume ratio of 100:1. The obtained mixture was sonicated using a probe with a tip diameter of 2 mm at 240 W and a constant frequency of 20 kHz in an ultrasonic homogenizer (JN92-IIDN, Xinzi Biotechnology Instrument Co., Ltd., Ningbo, China). The processing times were 5 min with pulse durations of 3 s on and 2 s off. The sonication treatment allowed CAPE incorporation into the RPNPs core during the self-assembly. Then, the sonication-treated samples were centrifuged at 1000× *g* for 10 min using a CENCE-L550 centrifuge (Cence Centrifuge Instrument Corp., Changsha, China) to remove the unencapsulated CAPE. The concentration of solubilized CAPE in the supernatant was determined using UV-Vis absorbance measurements at a wavelength of 323 nm. The encapsulation efficiency (EE) and loading capacity (LC) was determined according to the following equation:(1)$$\mathrm{EE}\;\left(\%\right)=\frac{\mathrm{amount}\;\mathrm{of}\;\mathrm{encapsulated}\;\mathrm{CAPE}}{\mathrm{total}\;\mathrm{amount}\;\mathrm{of}\;\mathrm{CAPE}}\times 100\%$$
(2)$$\mathrm{LC}\;\left(\%\right)=\frac{\mathrm{amount}\;\mathrm{of}\;\mathrm{encapsulated}\;\mathrm{CAPE}}{\mathrm{weight}\;\mathrm{of}\;\mathrm{sorghum}\;\mathrm{peptides}}\times 100\%$$

### 2.6. Storage Stability of CAPE-RPNPs

The CAPE-loaded RPNPs (CAPE-RPNPs) were dispersed in deionized water and then stored under 4 °C and 25 °C environments for 18 days. Aliquots of the CAPE-RPNPs suspensions at different time points (0, 1, 3, 7, 12, 18 day) were taken out to analyze particle size and CAPE concentration using the abovementioned methods. The storage stability of the CAPE-RPNPs was analyzed according to the change in particle size and remaining amount of CAPE.

### 2.7. Release Behavior of CAPE-RPNPs

It is important to understand the release behavior of the encapsulated compounds. The release profiles of CAPE-RPNPs were investigated using an in vitro gastrointestinal digestion model according to a previous study [15] with minor modification. Briefly, the simulated gastric fluid (SGF) and intestinal fluid (SIF) were prepared, respectively. The SGF contained 2.0 mg/mL pepsin and had a pH of 2.0. The SIF was composed of 2.0 mg/mL pancreatin, and its pH value was adjusted to 6.8. For the in vitro gastrointestinal digestion, CAPE-RPNPs were dispersed with SGF or SIF, and then were transferred into a 10-kDa dialysis bag. The mixture was gently shaken in a PBS solution, allowing the release of the encapsulated CAPE. An aliquot of buffer containing the released CAPE was taken out at 0, 15, 30, 60, 90 and 120 min, and was quantified by UV-Vis absorbance measurements at 323 nm. The following formula was used to calculate the cumulative CAPE release:(3)$$\mathrm{Cumulative}\;\mathrm{CAPE}\;\mathrm{release}\;\left(\%\right)=\frac{V_{0}C_{n}+V_{d}\sum_{1}^{n-i}C_{i}}{m_{i}}\times 100\%$$
where *V*_0_ is the volume of the release medium, *C_n_* (mg/mL) is the concentration of CAPE in release medium, *V_d_* is the volume of the medium taken out, *m_i_* (mg) is the initial CAPE amount in RPNPs.

### 2.8. Fluorescence Quenching Analysis

The interactions between CAPE and rice peptides were investigated using fluorescence spectroscopy. Briefly, different amounts of CAPE were dispersed in the solution of rice peptides. The final concentrations of CAPE were changed from 0 to 100 μmol/L (0, 5, 10, 20, 40, 60, 80, 100 μmol/L). Then, the mixtures were incubated and measured at 298 K and 310 K. The fluorescence spectrum was obtained using an fluorescence spectrophotometer (RF-5301PC, Shimadzu, Tokyo, Japan) with an excitation wavelength at 280 nm. The emission range was set with a range of 300 to 450 nm. The slit widths were set as 5 nm. The obtained fluorescence spectrum was further used to determine the Stern–Volmer quenching constant (K_sv_), quenching rate constant (K_q_), binding constant (K_a_), binding sites numbers (*n*), and thermodynamic parameters (ΔH, ΔS, ΔG).

### 2.9. Statistical Analysis

The values are expressed as the mean ± standard deviation (SD). All experiments were performed in at least three replicates. When *p* was <0.05 by analysis of variance and the Duncan test, difference was considered to be significant.

## 3. Results and Discussion

### 3.1. Effective Diameter, Polydispersity Index (PDI), and Internal Forces of Rice Peptide Nanoparticles (RPNs)

Rice protein has a high nutritional value because of its rich essential amino acids and proper amino acid pattern [8]. In addition to its use for bioactive peptides preparation [16], the application of rice protein in nano-delivery systems has received great interest of researchers [8,17]. However, poor water solubility limits its application in such fields [11]. Enzymolysis of proteins is a promising strategy to develop nano-delivery systems. Some researchers have reported that amphipathic peptides could be produced from natural proteins such as α-lactalbumin, egg yolk, and cruciferin by enzymolysis, thereby self-assembling into nanomicelles or nanoparticles [9,10,18]. In this study, rice protein was subjected to controllable enzymolysis using Alcalase^®^ 2.4 L. The self-assembled peptides were produced after hydrolysis with a degree of hydrolysis of around 18%. The particle size distribution and polydispersity index (PDI) of rice peptide nanoparticles (RPNs) is presented in Figure 1A. The result indicated that a rice peptide could self-assemble into nanoparticles with an effective diameter of about 210 nm. The particle size of RPNs was comparable to that of unfractionated cruciferin peptides nanoparticles [18], but was much bigger than that of peptides nanomicelles obtained from partial hydrolysis of α-lactalbumin and egg yolk [9,10]. It has been believed that amino acid sequences and preparation methods have a significant effect on peptide self-assembly [19], which may explain the differences in the size of different peptides nanoparticles. The PDI has been widely used as an indicator for evaluating the size distribution of particles. In general, the smaller the PDI, the narrower the particle size distribution. When PDI is more than 0.5, the system is polydisperse; if PDI is closer to zero, it shows the monodisperse system; and a PDI of less than 0.5 is considered optimum for nanoparticles [20]. Polydisperse systems have a greater tendency to aggregate than monodisperse systems. The PDI value of RPNs was about 0.23, suggesting that RPNs had a good dispersity.

We further investigated the internal forces in stabilizing the structure of RPNs. Denaturing agents including SDS, DTT, and urea have been widely used alone or in combination to analyze the interaction forces existing on the proteins or peptides nanoparticles. It is reported that SDS, DTT, and urea can disrupt the hydrophobic interaction, disulfide bond, and hydrogen bond, respectively [21]. The particle sizes of RPNs in different denaturing agents are represented in Figure 1B. When treated with a 0.5% SDS solution, the mean diameter of RPNs had no significant differences compared to that of RPNs in deionized water, suggesting that the hydrophobic interaction was not a main force to maintaining RPNs structure. In contrast, 30 mM DTT and 6 M urea alone or in combination significantly increased the mean diameter of PRNs (*p* < 0.05), implying that the disruption of disulfide bonds and hydrogen bonds could result in the expansion of RPNs. It is noteworthy that the increased degree of RPNs diameter treated by urea was greater than that of the RPNs diameter treated by DTT, indicating that hydrogen bonds exhibited a predominant role for maintaining the structure of RPNs. Glutelin has been reported to be a dominant protein fraction in rice proteins, and accounts for 79–83% of the protein in milled rice [22]. Disulfide bonds exerted an important role in the polymerization of the glutelin subunits and the formation of macromolecular complexes [22]. Consistent with the previous study, disulfide bonds also played prominent role in maintaining RPNs structure.

### 3.2. Encapsulation of CAPE by RPNs

The poor water solubility of CAPE (less than 10 μg/mL) greatly limits its efficacy and practical applications [23]. In this context, RPNs were utilized to encapsulate CAPE and improve its aqueous solubility. As shown in Figure 2, CAPE-loaded RPNs (CAPE-RPNs) had an effective diameter of about 240 nm, which was bigger than that of empty RPNs. Hence, it was speculated that self-assembled rice peptides encapsulated CAPE inside the hydrophobic core, resulting in an increase in nanoparticle size. The PDI of CAPE- RPNs was 0.22, suggesting that CAPE-RPNs had a good dispersity. The encapsulation efficiency (EE) and loading capacity (LC) were found to be 77.77 ± 0.35% and 3.89 ± 0.11%, respectively. The EE of CAPE-RPNs in this study is higher than that obtained by other researchers using skim milk microcapsules [1] and asiatic acid nanoparticles [7], but is lower than that of casein-sodium alginate nanoparticles [4]. In addition, the LC of CAPE-RPNs was lower than that of skim milk microcapsules [1]. Loading capacity depends mainly on the compound solubility in the matrix material, which might explain the difference of the nanocarriers in loading CAPE. Notably, the water solubility of CAPE was increased to 90 μg/mL, which was about 45-fold higher than free CAPE in water (about 2 μg/mL). These results suggested that RPNs could be used to improve the aqueous solubility of CAPE.

### 3.3. Storage Stability

In practical application, a nanocarrier should remain stable during the desirable shelf life. Hence, it is important to evaluate the storage stability of CAPE-RPNs. The CAPE-RPNs were stored at 4 °C and 25 °C for 18 days, and their size changes are shown in Figure 3A. At 25 °C, the mean diameter of CAPE-RPNs kept around 250–310 nm in the first 7 days, but their size was sharply increased to more than 3000 nm. This phenomenon can be attributed to the aggregate of CAPE released from damaged RPNs due to microbial contaminations. At 4 °C, the size of CAPE-RPNs during the whole storage period was always around 220 nm, suggesting that CAPE-RPNs had a good stability at 4 °C. The improved storage stability of CAPE-RPNs at a lower temperature was probably attributed to the significant decrease in molecular aggregation, oxidative degradation, and microbial contaminations [24]. Moreover, no obvious change, including visual flocculation or aggregation, was observed during storage time.

The effect of RPNs encapsulation on CAPE stability was also evaluated. Figure 3B illustrates the remaining quantity of CAPE within an 18-day storage period. At 4 °C, the residual quantity of both encapsulated and free CAPE was decreased with the extension of storage time. However, it is noteworthy that the remaining amount of encapsulated CAPE was always higher than that of free CAPE throughout the storage time, suggesting that the encapsulated CAPE had a better stability than free CAPE. On the 18th day, the remaining amounts of free CAPE and encapsulated CAPE were 46.4% and 65.3%, respectively. The higher stability of the encapsulated CAPE in RPNs could be attributed to the presence of rice peptides, forming an isolating layer that protected CAPE against destruction. It has been reported that CAPE is susceptible to oxidative degradation due to its good scavenging capability for radicals [25]. Encapsulated CAPE exhibited a better storage stability than free CAPE, which might be attributed to the fact that CAPE was encapsulated in the internal area or core of RPNs, isolating it from external oxygen and light, thus preventing CAPE from being oxidized. At 25 °C, a similar tendency was observed for the change in the remaining amounts of free and encapsulated CAPE, and the remaining amounts of free CAPE and encapsulated CAPE were 41.5% and 59.2%, respectively. It is noteworthy that the remaining amounts of CAPE at a lower temperature were always higher than that at a higher temperature, implying that a lower temperature could decrease the degradation of CAPE. Taken together, the above data indicated that RPNs could be used to increase the storage stability of CAPE.

### 3.4. In Vitro Release of CAPE-RPNs

It is necessary to study the release behavior of CAPE-RPNs during the digestion process, because their gastrointestinal digestive stability played an important role in protecting CAPE. The release characteristic of encapsulated CAPE was evaluated by measuring the cumulative CAPE-release percentage in the gastrointestinal digestion model in vitro, and the result is shown in Figure 4. In SGF, the cumulated CAPE release of CAPE-RPNs quickly increased to 49.0% within 30 min and remained constant afterward. The result was similar to the release of CAPE encapsulated by asiatic acid nanoparticles [7], but was significantly different compared to that of CAPE encapsulated by Ca-casein-sodium alginate nanoparticles [4]. In SIF, an initial burst release of CAPE was observed during the first 30 min, and a stable release plateau was reached after 60 min. After a simulation of intestinal digestion for 120 min, about 50% of CAPE was released. Compared with a previous study, which reported that 60% of Ca-casein-CAPE-NaAlg nanoparticles were released during SIF period [4], CAPE-RPNs displayed a lower release amount, suggesting that CAPE-RPNs had a more stable nanostructure, thereby better protecting CAPE against extreme environments during digestion.

### 3.5. Interaction Mechanism between RPNs and CAPE

#### 3.5.1. Fluorescence Quenching

The interaction mechanism between RPNs and CAPE was investigated by fluorescence quenching. This method has been widely used to study the interaction between proteins and small compounds. Due to the presence of fluorescence groups, including tryptophan and tyrosine, when protein or peptide molecules are excited at 280 nm, they can emit strong endogenous fluorescence at about 350 nm [26]. However, fluorescence quenching will take place if proteins or peptide molecules interact with non-fluorescence substance by excitation state reaction, complex formation, energy transferor, or collision quenching [27]. Some parameters including binding constants, number of binding sites, and thermodynamic parameters can be obtained from the analysis of fluorescence quenching spectra, thereby elucidating the interaction mechanism between RPNs and CAPE.

When RPNs were present with CAPE (its final concentration varied from 0 to 100 μmol/L), their fluorescence quenching spectra at 298 K and 310 K are shown in Figure 5A and Figure 5B, respectively. The results showed that CAPE could dose-dependently quench the fluorescence of RPNs, demonstrating that an interaction existed between RPNs and CAPE. Meanwhile, a successive red shift was also observed when the concentration of CAPE was increased. A greater hydrophobicity of the environment could lead to a higher fluorescence intensity, which was indicative of the changes in tryptophan and tyrosine of RPNs toward a more hydrophilic microenvironment. Consistent with this result, Wei et al. also reported that the interaction between CAPE and casein generated an environment with more hydrophily [4].

#### 3.5.2. Binding Constants and Number of Binding Sites

The mechanism of CAPE quenching fluorescence of RPNs was further analyzed. It has been reported that fluorescence quenching is divided into static quenching and dynamic quenching [28]. Dynamic quenching means physical collision or energy transfer between fluorescent molecules and quenching agents, while static quenching refers to the formation of complexes between ground state fluorescent molecules and quenching agents by weak bonds [28]. The value of static quenching constant decreases, while dynamic quenching constant increases when the temperature is increased [28]. The following Stern–Volmer equation could be used to determine the type of quenching mechanism by calculating the Stern–Volmer quenching constant (K_sv_):(4)$$\frac{F_{0}}{F}=K_{\mathrm{sv}}\left[Q\right]+1=K_{q}\tau_{0}\left[Q\right]+1$$
where F_0_ and F are the fluorescence intensities of RPNs when CAPE is absent or present, respectively. K_sv_ and K_q_ represent the Stern–Volmer quenching constant and quenching rate constant, respectively. [Q] and τ_0_ are the total concentration of CAPE and average fluorescence lifetime of RPNs (10^−8^ s), respectively.

The F_0_/F and [Q] in the Stern–Volmer plots obtained at 298 K and 310 K exhibited a linear correlation, as shown in Figure 5C. The K_sv_ values were obtained by calculating the line slope and are summarized in Table 1. It was found that the K_sv_ value decreased when the temperature was increased. This result indicated that a static quenching existed between RPNs and CAPE, leading to the formation of complexes. Based on the result, it is reasonable to believe that rice peptides encapsulated CAPE in the core by self-assembly, thus forming the CAPE-RPNs. We further calculated the number of binding sites (*n*) and binding constant (K_a_) between CAPE and RPNs with the following equation:(5)$$\mathrm{lg}\frac{F_{0}-F}{F}={\mathrm{lgK}}_{A}+n\mathrm{lg}\left[Q\right]$$

The double logarithmic curve of the interaction between RPNs and CAPE is presented in Figure 5D. It shows a significant positive linear correlation at both 298 K and 310 K. The K_a_ and *n*, which were calculated from the slope and intercept, respectively, are summarized in Table 1. It clearly shows that the values of K_a_ and *n* decreased with the increase in temperature, suggesting that the binding affinity of RPNs and CAPE was weakened when the temperature was increased.

#### 3.5.3. Thermodynamic Parameters

The following formulas were used to calculate the thermodynamic parameters, thereby determining the types of interaction force between RPNs and CAPE:(6)$${\mathrm{lnK}}_{A}=-\Delta H/\mathrm{RT}+\Delta S/R$$
(7)$$\Delta G=-{\mathrm{RTLnK}}_{A}$$
(8)ΔG=ΔH−TΔS
where ΔG is Gibbs free energy change, ΔH is enthalpy change, ΔS is entropy change, and R is gas molar constant, 8.314 J·mol^−1^·K^−1^.

Table 1 summarizes the data of the thermodynamic parameters. It has been widely reported that four types of interaction forces exist between proteins/peptides and small molecules, including hydrophobic interactions, hydrogen bonds, electrostatic interactions, and van der Waals forces [29]. Thermodynamic parameters could be used to determine the type of interaction force. When ΔH < 0 and ΔS < 0, hydrogen bond and van der Waals force are predominant forces; when ΔH > 0 and ΔS > 0, hydrophobic interaction is the main force; when ΔH < 0, ΔS > 0, the electrostatic interactions are the main forces [30]. As shown in Table 1, both ΔH and ΔS are negative values, indicating that hydrogen bond and van der Waals force are the predominant forces between RPNs and CAPE. CAPE structurally contains two hydroxyl groups, which might explain the interaction forces between RPNs and CAPE, and the formation of CAPE-RPNs because these phenolic hydroxyl groups could form hydrogen bonds with amino acids with hydroxyl groups in RPNs [31]. Consistent with our study, Li et al. also reported that the hydrogen bonds and van der Waals forces were the main interaction forces between CAPE and human serum albumin [32]. However, other researchers found that the main interaction between CAPE micellar casein was hydrophobic force [33]. In addition, we found that the formation of CAPE-RPNs occurred spontaneously, which was demonstrated through the negative value of ΔG. Meanwhile, the absolute value of ΔG indicated that CAPE-RPNs formation was easier at the temperature of 298 K, which agreed with the result as reflected by K_a_ and *n*.

## 4. Conclusions

In conclusion, rice peptides were used for CAPE encapsulation to enhance the solubility and stability of CAPE in this study. Rice peptides could self-assemble into nanoparticles (RPNs) with a diameter of around 210 nm with disulfide bonds and hydrogen bonds. RPNs exhibited a high encapsulation efficiency (77.77%) and loading capacity (3.89%) for CAPE, and greatly increase the water solubility (45 times) and storage stability (1.4-fold) of CAPE. The interaction between RPNs and CAPE was mainly driven by van der Waals forces and hydrogen bonds. RPNs were a promising delivery system to improve the water solubility and stability of CAPE.

## Figures and Tables

**Figure 1 foods-13-00755-f001:**
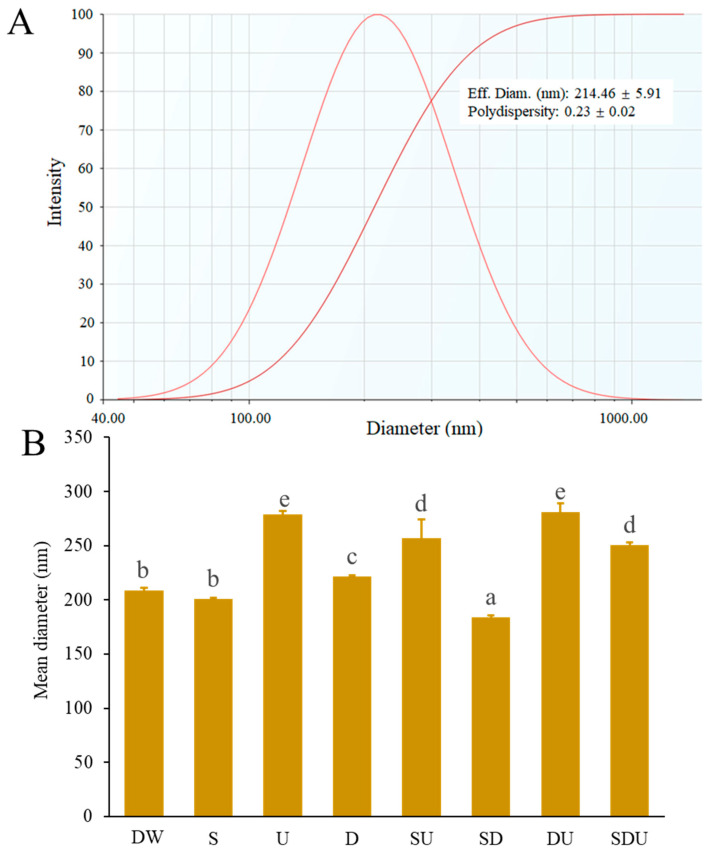
(**A**) Particle size distribution of rice peptides nanoparticles (RPNs). (**B**) Effects of different denaturing agents on the mean diameter of RPNs, DW: deionized water; S: SDS; U: Urea; D: DTT; SU: SDS + Urea; SD: SDS + DTT; DU: DTT + Urea; SDU: SDS + DTT + Urea, different letters suggest significant differences (*p* < 0.05).

**Figure 2 foods-13-00755-f002:**
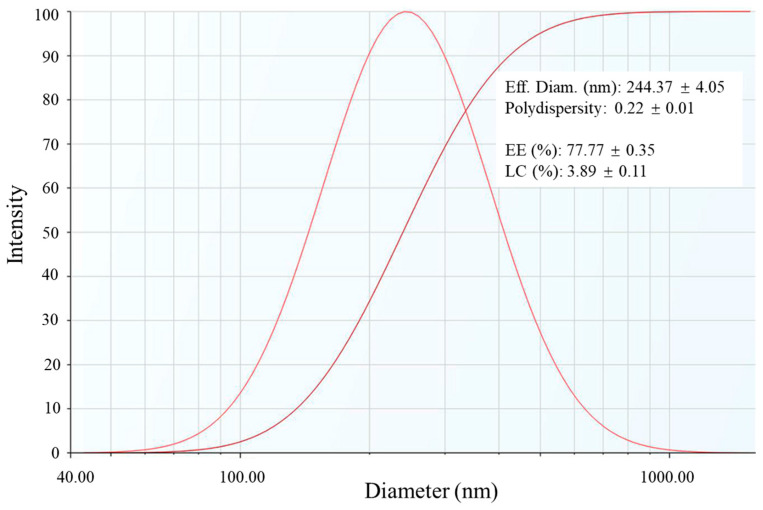
Particle size distribution of CAPE-loaded rice peptides nanoparticles (CAPE-RPNs).

**Figure 3 foods-13-00755-f003:**
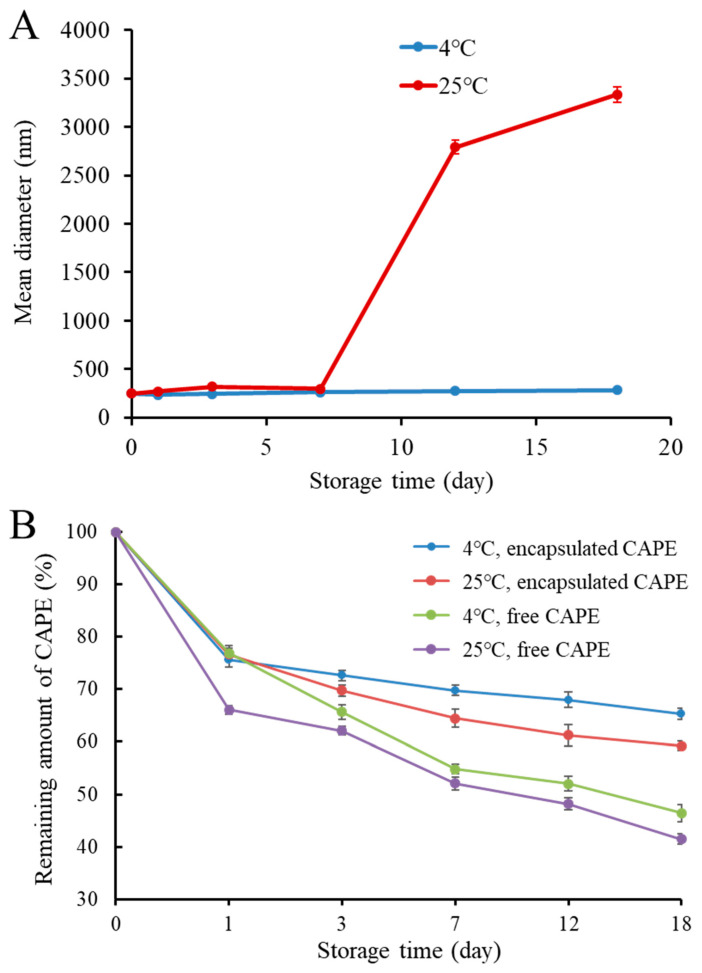
(**A**) Effect of storage time on mean diameter of CAPE-loaded rice peptides nanoparticles (CAPE-RPNs). (**B**) Relative remaining amount of free and encapsulated CAPE at 4 °C and 25 °C.

**Figure 4 foods-13-00755-f004:**
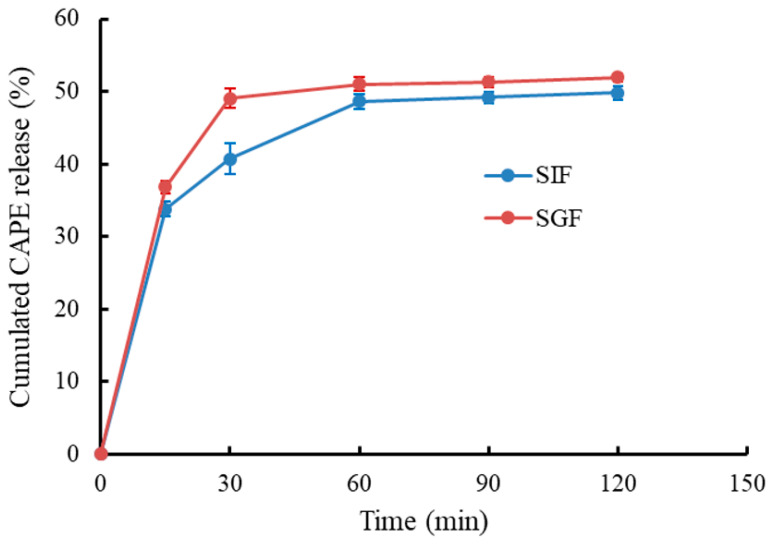
In vitro release behavior of CAPE-loaded rice peptides nanoparticles (CAPE-RPNs) in simulated gastric fluid (SGF) and intestinal fluid (SIF).

**Figure 5 foods-13-00755-f005:**
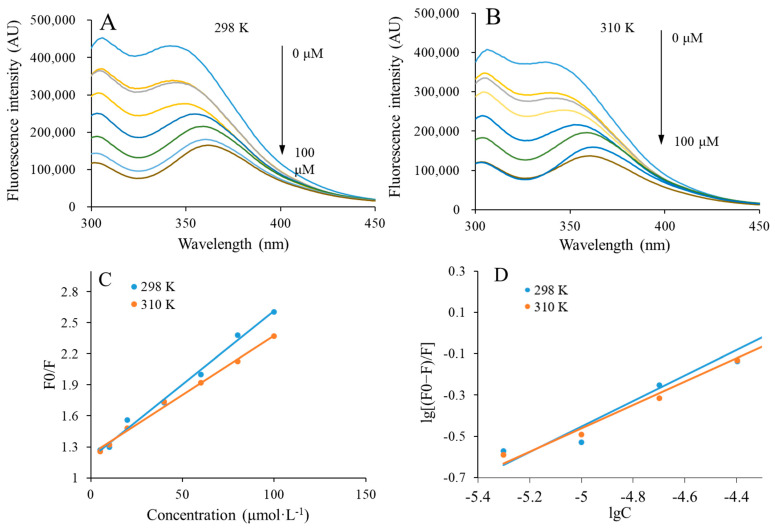
(**A**,**B**), Fluorescence quenching spectra of rice peptides nanoparticles (RPNs) in the presence of CAPE at 298 K and 310 K, respectively. (**C**) Stern−Volmer plot for the interaction of RPNs and CAPE at 298 K and 310 K. (**D**) The log [(F0 − F)/F] and log C double logarithmic plots for the interaction of RPNs and CAPE at 298 K and 310.

**Table 1 foods-13-00755-t001:** The Stern–Volmer quenching constant (K_sv_), quenching rate constant (K_q_), binding constant (K_a_), binding sites numbers (*n*), and thermodynamic parameters (ΔH, ΔS, ΔG) for CAPE interaction with rice peptides (RPs) at 298 K and 310 K.

Temperature (K)	K_sv_ (L∙mol^−1^)	K_q_ (L∙mol^−1^∙s^−1^)	Quenching Type	K_a_ (L∙mol^−1^)	*n*	ΔG (kJ∙mol^−1^)	ΔH (kJ∙mol^−1^)	ΔS (J∙mol^−1^∙K^−1^)
298	1.42 × 10^4^	1.42 × 10^12^	static quenching	0.44 × 10^3^	0.62	−20.79	−41.65	−70
310	1.15 × 10^4^	1.15 × 10^12^	static quenching	0.23 × 10^3^	0.56	−19.95	−41.65	−70

## Data Availability

The original contributions presented in the study are included in the article, further inquiries can be directed to the corresponding author.

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
