# Peer review of "Caffeic Acid Phenethyl Ester Encapsulated in Self-Assemble Rice Peptides Nanoparticles: Storage Stability, In Vitro Release, and Their Interaction Mechanisms"

_foods, 2024, doi:10.3390/foods13050755_

Round 1
Reviewer 1 Report
Comments and Suggestions for Authors
Reviewer 2 Report
Comments and Suggestions for Authors
There are several typing and formal errors. Please, revise the MN, like in line 35 of page 1, line 53 of page 2, and several other ones.
Page 2 line 79 the name of organisms, including microorganisms should be in italic.
Hexane is normal-hexane or cyclohexane?
In page 2, about the sentence The pH of obtained supernatant was adjusted to 5 and then 91 was centrifugated at 3000 × g for 10 min. Report how the pH was adjusted. About the sentence The protein purity of rice protein was about 92%. Report how the purity was obtained, also with the appropriate reference.
Page 4 about the sentence Rice protein is a valuable plant protein because of its rich essential amino acids and 169 proper amino acid pattern. In addition to the use for bioactive peptides preparation, the application of rice protein in nano-delivery systems has received great interest of researchers. However, poor water solubility limits its application in such field. Enzymolysis of proteins is a promising strategy to develop nano-delivery systems. Add some references.
About the sentence In this study, rice protein was subjected to controllable enzymolysis by Alcalase® 2.4L. The self-assembled peptides were produced after hydrolysis with a degree of hydrolysis of around 18%. Consider that this information was already reported.
Page 6 change Asiatic acid nanoparticles in asiatic acid nanoparticles and the same in page 8
Page 9 about the sentence Due to the presence of fluorescence groups including tryptophan and tyrosine, when protein or peptide molecules are excited at 280 nm, they can emit strong endogenous fluorescence at about 350 nm[23] why phenylalanine was not considered?
About Conclusions. How we can be sure that CAPE maintains its properties, as reported in the Introduction, after the microincapsulation?
Comments on the Quality of English Language
It is necessary to revise the English and the meaning, like in the sentence Rice protein is a valuable plant protein because of its rich essential amino acids and proper amino acid pattern.
Round 2
Reviewer 1 Report
Comments and Suggestions for Authors
Dear authors,
The second MS version is definitely much better.
However, I strongly encourage you to use an anti-plagiarism tool to avoid a high percentage of plagiarism (Sections 2.2, 2.4 and 3.5.3. are abundant).
You should check them to have an original content.
There are also tools for text polishing and I recommend you to use them instead of pasting old information on a new template.
Kind regards,

Author Response
Response: Thank you very much for your comments on our manuscript. These comments are very helpful for revising this manuscript. According to the plagiarism report, we have revised the manuscript carefully. Please pay attention to the sections 2.2, 2.4, 3.3, 3.5.1, 3.5.3 and other sections in the revised manuscript.